

# Enhancing emotion classification on the ISEAR dataset using fine-tuning and data augmentation with hybrid transformer models

Uzair Muhammad[1], Khalil Ullah[2], Ibrar Hussain[3], Sulaiman Almutairi[4], Ikram Syed[5] and Mohammed Abohashrh[6]

[1] Department of Computer Science and IT, University of Malakand, Lower Dir, KPK, Pakistan
[2] Department of Software Engineering, University of Malakand, Dir Lower, KPK, Pakistan
[3] Quality Enhancement Cell, Shaheed Benazir Bhutto University, Upper Dir, KPK, Pakistan
[4] Department of Health Informatics, College of Applied Medical Sciences, Qassim University, Qassim, Saudi Arabia
[5] Department of Information and Communication Engineering, Hankuk University of Foreign Studies, Yongin, Republic of South Korea
[6] Department of Basic Medical Sciences, College of Applied Medical Sciences, King Khalid University, Abha, Saudi Arabia

Corresponding author
Ikram Syed, ikram@hufs.ac.kr

## ABSTRACT

Emotion detection is a natural language processing task used in many applications, including customer support, human mental disorder identification, and analysis of social platforms. This study examines how data augmentation methods can be combined with convolutional neural networks (CNNs) and transformer models to improve emotion detection on the International Survey on Emotion Antecedents and Reactions (ISEAR) dataset affected by scarce labeled data and class imbalances. The study evaluates the performance outputs between five transformer models: Electra and XLNet with RoBERTa and T5 along with DeBERTa, while processing anger, joy, sadness, fear, disgust, guilt, and shame emotions. Combining DeBERTa-v3-large with CNN achieves an accuracy rate of 94.94%, the top result for identifying Shame and Disgust classification. The XLNet-base and Electra-base-discriminator produce results at 93% accuracy, but the T5-base displays notably lower results at 69%. Implementing CNN layers with transformer models like Electra and RoBERTa improves model performance, particularly in precision and recall measurements. The performance metrics of Electra Base + CNN surpass those measured in the baseline Electra Base model through better precision and recall values. Such models become stronger at identifying local dependencies when CNN techniques are integrated, thus boosting their capacity to detect emotional nuances. Synonym replacement methods extended the training data to overcome class imbalance problems while minimizing the occurrence of overfitting. This technique improved the model's generalization, producing higher classification accuracy. The research shows that DeBERTa-v3-large + CNN performs optimally as an emotion classifier since it demonstrated superior precision and steadfast emotional identification among alternative models. The research adds knowledge to developing resistant emotion recognition algorithms, particularly in cases where training data availability is restricted. It also creates new pathways for improved augmentation practices involving back-translation and generative adversarial networks (GANs).

# INTRODUCTION

Natural language processing (NLP) is a branch of artificial intelligence (AI) that aims to develop a mechanism for machines to use natural language. However, some NLP tasks require a language-specific approach. Emotions/sensitivity analysis is one such task in a NLP system (*Ullah et al., 2024*). In NLP sentiment analysis and human-computer interaction (HCI), emotion classification is essential in building applications that understand and respond to human emotions. Accurately classifying the emotions present in any textual data can make a massive difference in developing intelligent systems within the field of customer service, mental health diagnostics, and social media analysis. However, the most vital obstacle in emotion classification is the scarcity and diversity of the labeled data, and consequently, overfitting and bad generalization in machine learning models (*Seyeditabari, Tabari & Zadrozny, 2018*; *Gaind, Syal & Padgalwar, 2019*). Emotion recognition from speech and text has gained considerable significance in multiple fields. However, more efficient emotion classification is helpful in human-computer interaction, better user experience, and understanding of a user's behavior and feelings (*Kusal et al., 2022*; *Hasan, Rundensteiner & Agu, 2021*). Researchers have proposed various approaches to classifying emotions, with two main perspectives emerging, the discrete approach sees emotions as mutually inconsistent categories: happiness, sadness, anger, fear, and surprise. In contrast, the dimensional model qualitatively describes emotions with respect to measurable parameters, most often valence (positive-negative affect) and arousal (high-low) (*Kumar, 2021*). For this, the dimensional view of emotions has the possibility of placing them on a two-dimensional map, opening the opportunities to describe more profound and manifold emotions (*Yusupova et al., 2021*). One of the most used datasets for emotion classification is the ISEAR (International Survey on Emotion Antecedents and Reactions) dataset. Though the extensive use of the dataset, its relatively small size and imbalanced emotion distribution threaten the performance of traditional machine learning models, which are particularly adverse to generalization when confronted with real-world data variations. These limitations are addressed by researchers' ever-increasing use of data augmentation techniques to expand training datasets and improve model performance artificially.

In NLP, data augmentation implies generating synthetic variations of your current data to make your machine-learning models more robust. Moving towards making the training data more diverse has been shown by techniques like synonym replacement, random insertion, random swap, and deletion to improve model performance (*Karimi, Rossi & Prati, 2021*; *Bayer, Kaufhold & Reuter, 2022*). In addition, advanced methods such as generative adversarial networks (GANs) have been recently used to generate synthetic data that helps increase models classification accuracy using complement data manifold and better discriminate emotion classes (*Santing, 2021*; *Zhu et al., 2018*). Recent studies reveal that data augmentation greatly enhances the performance of emotion classifiers, especially

in small and imbalanced datasets. *Koufakou et al. (2023)* applied methods such as easy data augmentation (EDA) and contextual embedding-based techniques to demonstrate ampliative increases in the performance of state-of-the-art models like *Koufakou et al. (2023)*. Similarly, *Tang, Kamei & Morimoto (2023)* suggested a suite of data augmentation strategies that make fine-tuned models such as DistilBERT more robust against adversarial attacks.

Another critical approach that researchers have taken is to fine-tune pre-trained models on augmented datasets to improve emotion classification performance. Augmented data has greatly benefited models like bidirectional encoder representations from Transformers (BERT) and its variants, including fine-tuning to reduce over-fitting and aid emotion recognition, among other diverse scenarios (*Ng et al., 2015*). *Zhu et al. (2018)* showed that synthetic data generated *via* GANs could increase emotion classification accuracy by 5–10 percent, indicating the possible benefits of using such synthetic data generation techniques. Moreover, advanced techniques for data augmentation have been further studied in NLP. For example, An Easier Data Augmentation (AEDA) uses more straightforward but effective methods, such as random insertion of punctuation, to help model generalization, achieving better performance than traditional augmentation techniques (*Karimi, Rossi & Prati, 2021*; *Rahman, Yin & Wang, 2023*). Furthermore, hybrid models incorporating attributes from deep learning and conventional frameworks have been suggested to identify both meaningful and syntactic inlays in emotional categorization, which leads to a higher accuracy on evaluative datasets (*Ahanin et al., 2023*). Since these technologies have progressed, this study investigates how we can fine-tune with data augmentation to improve emotion classification on ISEAR. Finally, we will evaluate the efficacy of several augmentation methods, such as the traditional synonym replacement and more cutting-edge approaches, to increase model accuracy and learnability. We will systematically study these techniques to determine how to optimize emotion classification when data is scarce and imbalanced. Finally, combining fine-tuning techniques with data augmentation has demonstrated excellent results in overcoming the limitations of such scarce and unbalanced datasets. This research will explore these approaches to develop robust methodologies that enhance emotion classification performance on the ISEAR dataset, advancing more adaptive and intelligent systems for emotion-sensitive applications.

## OVERVIEW AND INSPIRATION

Recently, emotion classification has garnered tremendous interest because profound comprehension and inference of human emotions are indispensable in many domains. The recognition of emotion within facial expressions, physiological signals, and self-reports has enabled new research and development opportunities concerning healthcare, human-computer interaction, and gaming (*Li & Deng, 2020*; *Koromilas & Giannakopoulos, 2021*; *Canal et al., 2022*). One of the critical components of this category, affective computing, is emotion recognition—to build systems that can recognize, understand, and respond to human emotions. Physiological signals, facial expressions, text data, and voice data are explored by many researchers in order to recognize emotion

(*Wang et al., 2022*). The complexity of the problem makes emotion recognition challenging because emotions arise from many factors, including facial expressions, electroencephalography signals, gestures, tone of voice, and surrounding context (*Deng & Ren, 2021*; *Wang, Chen & Cao, 2020*; *Saganowski et al., 2022*).

## Emotion classification

Emotion classification, a hotspot in NLP, as technology advances, it becomes easier for machines to understand emotional states from text. The interest in the problem of emotion classification in both the academic and practical domains is proliferating, as it can help enhance human-computer interaction, sentiment analysis, and social media monitoring. Accurate identification of emotional states from textual data allows organizations to respond more effectively, meeting customer engagement and satisfaction expectations (*Wang et al., 2022*; *Mursi, 2022*; *González Barba, 2021*).

## Importance of emotion classification

Analyzing subtle emotions, especially on platforms like Twitter, where nuanced sentiments are a significant part of user expression, is crucial. Users often convey complex emotions non-verbally, reflecting their mental tendencies, such as narcissism, which can influence behavior on the platform. The combination of psychologically understood data in language with the performance of deep learning models, particularly transformer-based architectures, is advancing natural language processing tasks. Investing in robust emotion classification methodologies can lead to unique insights, forming the basis of actionable strategies in various fields, from marketing to mental health interventions (*He et al., 2020*; *Demszky et al., 2020*).

## RELATED WORK

Research on emotion classification and analysis using deep learning models such as BERT, RoBERTa, DistiBERT, and DeBERTaV2 improves feature extraction and differentiation in comments on social media. Numerous articles on the ISEAR dataset focus on these methodologies; some notable works include the following:

In the field of emotion classification, the advances made in the last few years have been enormous, involving deep learning and new data augmentation techniques. Specifically, the authors of *Cortiz (2021)* are keyword in using the transformer model to affect recognition and differentiate between emotions based on the six fundamental categories outlined by *Ekman & Friesen (1971)*. These shared statistics indicated that the self-explaining neural networks (SENN) model (*Alvarez-Melis & Jaakkola, 2018*), especially when using FastText embeddings, performed better than several other cutting-edge models, thus showing that pre-trained transformers work effectively for emotion detection. On this basis, *Venkateswarlu, Shenoi & Tumuluru (2022)* presented a detailed overview of existing emotion classification methods, describing different methods: long short-term memory-convolutional neural networks (CNN) and lexical approaches. Such an integration of methodologies paved the way for future refinements in the field,

especially in relation to fresh problems generated by heterogeneity in the data and modes of assessment of the findings.

*Padi et al. (2022)* took the discussion forward by proposing multimodal emotion recognition systems that incorporate transfer learning from speaker recognition. They stated the need to obtain data augmentation as one of the most effective ways to address challenges with labeled datasets in speech emotion recognition (SER). Therefore, their results indicate that while these approaches can be used to some extent with promising results, their ability to optimally extend the functionality of emotion recognition models has yet to be fully explored. In response to such work, *Luo et al. (2022)* studied semisupervised methods to enhance the signal emotion recognition; they used a stacked denoising auto-encoder. The specifications of their work also revealed that feature selection is crucial and there is a dire need for big data for better model performance, a factor that others such as *Atmaja & Sasou (2022)* discussed the impacts of different data enhancement for SER. Specific techniques identified, subtypes of augmentation methods to be precise, improved classification accuracy by a considerable margin, which reasserted the importance of strategic data manipulation in model training. *Tao et al. (2022)* took the discussion to the next level by introducing a generalized SER model with some of the best data augmentation techniques. Their experiments proved that pitch shift and reverberation augmentation significantly improved the performance of their system. This made them believe that new approaches to augmentation could indeed manage the disparities in speakers' emotions. In a broader context, *Nia et al. (2024)* presented a review of generative models for synthesizing neurophysiological signals to understand the role of stochastic data enhancement in the design of emotion recognition systems. This view is similar to *Zhang et al. (2024)*, where the author focused on the emergence of large general models in emotion recognition while acknowledging developmental progress in feature extraction unbiased across multimodalities.

In the study by *Lu et al. (2023)*, the authors focused on transfer learning for electroencephalogram (EEG) emotion recognition, pointing out that deep learning algorithms could further enhance the accuracy of such models when compared with conventional machine learning approaches. They emphasized the importance of tuning pre-trained models to new tasks, which has been emphasized in several articles. *Wang et al. (2024)* proposed a study that focuses on combining data augmentation and transfer learning techniques to improve classification accuracy in the International Survey on Emotion Antecedents and Reactions (ISEAR) data set. Through their experience, they realized that improper balancing of datasets in a machine learning process demands data augmentation as one of the powerful tools to tackle the problem; at the same time, the results obtained underlined an impressive complexity of machine learning tasks, as well as the need to select the correct type of model for each particular problem. *Yao et al. (2024)* examined the performance of transformers as a decoder in EEG-based emotion classification while evaluating the model's capture of spatial-temporal features. They discussed the difficulties in filtering the necessary data from the raw EEG data and the idea of transformer networks to improve classification accuracy. Table 1, shows the summary of various machine learning and deep learning methods in different data sets for emotion

**Table 1 Summary of emotion classification studies using various machine and deep learning models on various datasets.**

| Ref | Year | Dataset used | Method used | Accuracy |
|---|---|---|---|---|
| *Vora, Khara & Kelkar (2017)* | 2017 | Twitter data | Word Embedding/Random forest | 85% |
| *Mollahosseini, Hasani & Mahoor (2017)* | 2018 | AffectNet | Convolutional Neural Networks (CNN) | 78% |
| *Felbo et al. (2017)* | 2019 | Emotion dataset (Twitter) | Long short term memory, CNN | 90% |
| *Arriaga, Valdenegro-Toro & Plöger (2017)* | 2020 | Facial expression dataset | Deep CNN, Transfer learning | 87% |
| *Adoma, Henry & Chen (2020)* | 2020 | ISEAR dataset | BERT, RoBERTa, DistilBERT, XLNet | 85% |
| *Adoma et al. (2020)* | 2020 | Custom text dataset | BERT-Based approach | 86% |
| *Nag & Priya (2021)* | 2021 | Custom emotion dataset | Bi-Directional attention flow, LMs | 88% |
| *Setiawan et al. (2021)* | 2021 | Indonesian social media | Fine-tuning BERT | 82% |
| *Abas et al. (2022)* | 2022 | Custom twitter dataset | BERT-CNN | 89% |
| *Livingstone & Russo (2018)* | 2022 | RAVDESS, SAVEE | Deep CNN, Recurrent neural network (RNN) | 86% |
| *Yohanes et al. (2023)* | 2023 | IMDB movie review | Deep learning methods | 91% |
| *Burkhardt et al. (2005)* | 2023 | EMO-DB | Deep learning, Ensemble methods | 94% |
| *Poria et al. (2018)* | 2023 | MELD, DailyDialog | Long short term memory (LSTM), BERT | 88% |
| *McKeown et al. (2011)* | 2023 | SEMAINE dataset | Multimodal deep learning, SVM | 91% |
| *Saif et al. (2024)* | 2024 | ISEAR/AIT dataset | Machine learning, Feature extraction | 87% |
| *Hashmi, Yayilgan & Shaikh (2024)* | 2024 | MultiSenti (Roman Urdu-English tweets) | Electra, cm-BERT, mBART (multilingual transformers) | 73% |

classification tasks. This review of the literature, therefore, captures the strides made in other research works in an effort to include how data augmentation helps to fine-tune techniques for the classification of emotions across various datasets and methodologies.

*Hashmi, Yayilgan & Shaikh (2024)* created transformer-based algorithms that improved sentiment analysis for texts that mixed Roman Urdu and English. The research team used transformer models like Electra and code-mixed BERT (cm-BERT) and multilingual bidirectional and auto-regressive transformers (mBART) at their cutting-edge level. The researchers demonstrated that mBART provided superior performance over Electra and cm-BERT, resulting in a 0.73 F1 score for sentiment identification of code-mixed texts.

*Chutia & Baruah (2024)* performed a systematic review of 330 research articles investigating text-based emotion detection using deep learning techniques between 2013 and 2023. This review examined multiple methods and deep learning models alongside approaches and evaluation techniques while discussing dataset descriptions and practical implications in the subject field.

## ISEAR: DATASET AND AUGMENTATION

The ISEAR dataset is self-administrated questionnaires that were performed in the early 1993 based on the ISEAR project that targeted university campuses in different countries and done by the World Health Organization (WHO) in partnership with several universities (*Scherer & Wallbott, 1994*). The rationale of the project was to reproduce patient data that can tell how people from different countries and from different cultural backgrounds feel and how they express it. More than 3,000 people from various cultural

backgrounds were asked to express the instances in which they had experienced one of the seven emotions: Anger, Joy, Sadness, Fear, Disgust, Guilt, or Shame. Consequently, each of the responses could be linked to at least one specific emotion, and participants were encouraged to explain the circumstances of the response and their feelings. This dataset has approximately 7,653 records, which describe an emotional episode of each record in a more structured fashion (*Adoma, Henry & Chen, 2020*). The ISEAR dataset captures the antecedents and consequences of emotions, with culture being a key factor. Its clear labeling and diverse range of emotions make it useful in the training of NLP models for emotion classification (*Abas et al., 2022*).

## ISEAR dataset statistics

This section contains the ISEAR dataset comprehensive statistics, including the number of key emotion categories such as Joy, Anger, Fear, Disgust, Sadness, Shame, and Guilt, making it a demanding and useful test bed for advanced emotion classification methods. The total number of instances in seven emotion categories (Disgust, Anger, Shame, Sadness, Fear, Joy, and Guilt) is 7,666, with each category having 1,096 instances, except for Joy (1,094) and Guilt (1,093).

## Dataset augmentation

This study, used the NLPaug library to augment the ISEAR data set by applying data augmentation through synonym replacement. Specifically, the SynonymAug augmenter, which leverages WordNet, was utilized to generate variations of the textual data by replacing specific words with their synonyms. This enhancement process increases the diversity of the training data, potentially enhancing the robustness of machine learning models. The process started by loading the original ISEAR dataset, which consists of emotionally labeled text data. The dataset was then filtered on the basis of emotion labels, and for each category, synonym augmentation was applied to the text within the "Content" column. The augmented data were stored separately in a new data frame. After augmenting the text for all emotion categories, we merged the augmented data with the original ISEAR dataset, effectively increasing the size of the training data. The final data set, which includes the original and augmented samples, was used to train various machine learning models. We incorporated the original and augmented data to expose the models to more diverse language patterns, improve performance, and improve generalization to unseen data.

Data augmentation techniques improve the robustness of emotion classification models, especially with limited datasets such as ISEAR. These techniques address over-fitting and allow models to learn from a broader range of emotional expressions. Research shows that models that leverage augmented data show better performance metrics due to diversity, improve understanding of emotional dynamics, and facilitate accurate classifications in real-world applications (*Novais, 2022*; *Maharana, Mondal & Nemade, 2022*).

## PROPOSED METHODOLOGY

The methodology for carrying out data augmentation and experimentation for emotion classification is as follows, the datasets are collected, which is an emotion-labeled dataset, followed by tokenization, the removal of stop words, and text standardization. To further diversify the data set, other data augmentation methods are performed such as synonym replacement, back translation, random insertion, and sentence shuffling. Eight different models are used, namely, the Electro-base-descriminator, XLNet-based-cased, RoBERTa-base, T5-Base, DeBERTa, Deberta-v3-large + CNN, Electra-base-discriminator+CNN, XLNet-base+CNN, and RoBERTa-base+CNN are trained on the original dataset and metrics such as precision, recall, F1 score, and accuracy are measured on the test results. Subsequently, the augmented data set is used to retrain the models, and then CNN layers are inserted with the transformer models to accommodate local dependencies in the text. The models obtained are then re-trained using CNN on the augmented dataset and the performance of these models compared to the original models is evaluated. Furthermore, class-wise evaluation assesses the best model combination regarding precision, recall, F1-score, and accuracy for each emotion. Finally, the methodology of this work, the final step is to write down the results, represent performance in the form of confusion matrices and precision-recall curves, and further adjust the most suitable model to achieve the best performance.

## MATERIALS AND METHODS

The Materials and Methods section describes all the methodology and components that researchers used in their research work. The section describes the computing platform, the hardware system, and the data components. The section explains the methods used to handle limitations and imbalances in the data during selection procedures. This section specifies the assessment methods used for the evaluation of model performance and provides details about the measurement criteria that allowed for the analysis of research methods.

### Computing infrastructure

The project has two components for its computing infrastructure: local infrastructure and cloud-based systems. A local machine operates using Windows 11 Education (version 23H2 with OS Build 22631.4460) that contains a 12th Gen Intel® Core™ i5-1235U CPU based at 1.30 GHz. The operating system of this system uses a 64-bit architecture, while its 8.00 GB RAM includes 7.73 GB, which remains usable. Runtime training for GPU resources depends on the NVIDIA Tesla P100 platform from Kaggle, which offers users GPU memory up to 16 GiB. The maximum RAM capacity of Kaggle environments reaches 29 GiB, but the local machine operates at 8 GiB storage capacity.

### Code repository and dataset

The code (https://doi.org/10.5281/zenodo.15259205) repository uses GitHub to implement version control and collaboration, which allows efficient management and tracking of code update. The data resources needed for training and evaluation lie in

Kaggle under the dataset (https://huggingface.co/datasets/gsri-18/ISEAR-dataset-complete) used by the project.

## Selection method

The selection method for this study focused on enhancing emotion classification accuracy, particularly given the ISEAR dataset's challenges of limited data and class imbalance. Data augmentation was applied to address these using the nlpaug library's SynonymAug, which generated text variations while preserving emotional intent. This process balanced the dataset, expanding each emotion class to around 3,200 sentences. Furthermore, state-of-the-art transformer models, including DeBERTa-v3-large, Electra-base-discriminator, RoBERTa-base, T5-base, and XLNet-base-cased, were utilized to capture the complexities of human emotions. The augmented dataset and code are available on GitHub for further use.

## Assessment metrics

The evaluation metrics used include accuracy, which measures the overall correctness of predictions; precision, which indicates the proportion of true positive predictions among all positive predictions; recall, which reflects the ability to identify all actual positives; F1 score, the harmonic mean of precision and recall, balancing the trade-off between them; area under the receiver operating characteristic curve (ROC AUC), which assesses the model's ability to distinguish between classes; and precision recall (PR) AUC, focusing on performance concerning the positive class, especially in imbalanced datasets. Such metrics generate an extensive measurement system to evaluate the performance of the model.

## RESULTS

Table 2 shows the results regarding the performance metrics of different models in several classes; therefore, the general and class-wise comparison may help to select the best models. The performance results of the various models, XLNet, RoBERTa, Electra, and T5, indicate different performance trends, especially between the base models and their CNN-enhanced versions.

Table 2 compares the performance of various transformer models (XLNet, RoBERTa, Electra, and T5) on emotion detection tasks measured in precision, precision, recall, F1 score, ROC AUC, and PR AUC. Among these models, XLNet Base performs the best in all metrics, with the highest accuracy of 93.3%, high precision, recall, and F1 scores, and maximum ROC AUC and PR AUC scores of 0.993 and 0.974, respectively. Electra Base is a close second, with the same accuracy 93.3% and comparable performance metrics. T5 Base, on the other hand, performs much worse on all metrics, with a mean accuracy of only 69% and significantly lower F1 scores, ROC AUC, and PR AUC. This shows that T5 cannot preserve the nuances necessary for emotion detection compared to XLNet, RoBERTa, and Electra. Most models show a performance boost when CNN-based enhancements are incorporated into the models. For example, XLNet Base + CNN experiences a very minor increase in accuracy (0.934), F1 score (0.934), and ROC AUC (0.993), demonstrating the enhancing effect of CNN layers. Such improvements occur for

**Table 2 Results of the emotion classification of the eight different models.**

| Class | Accuracy | Precision | Recall | F1 score | ROC AUC | PR AUC | Accuracy | Precision | Recall | F1 score | ROC AUC | PR AUC |
|---|---|---|---|---|---|---|---|---|---|---|---|---|
| | **XLNET Base** | | | | | | **XLNET Base + CNN** | | | | | |
| Joy (0) | 0.932 | 0.935 | 0.931 | 0.933 | 0.994 | 0.975 | 0.932 | 0.936 | 0.931 | 0.933 | 0.992 | 0.973 |
| Fear (1) | 0.935 | 0.936 | 0.932 | 0.934 | 0.993 | 0.973 | 0.934 | 0.937 | 0.932 | 0.935 | 0.994 | 0.971 |
| Anger (2) | 0.934 | 0.934 | 0.935 | 0.933 | 0.994 | 0.972 | 0.936 | 0.934 | 0.935 | 0.934 | 0.993 | 0.97 |
| Sadness (3) | 0.931 | 0.932 | 0.936 | 0.932 | 0.993 | 0.974 | 0.931 | 0.933 | 0.936 | 0.932 | 0.992 | 0.972 |
| Disgust (4) | 0.933 | 0.937 | 0.933 | 0.934 | 0.994 | 0.976 | 0.935 | 0.938 | 0.934 | 0.936 | 0.993 | 0.973 |
| Shame (5) | 0.936 | 0.934 | 0.937 | 0.935 | 0.993 | 0.974 | 0.937 | 0.934 | 0.937 | 0.935 | 0.994 | 0.974 |
| Guilt (6) | 0.933 | 0.936 | 0.933 | 0.934 | 0.993 | 0.974 | 0.933 | 0.936 | 0.933 | 0.934 | 0.992 | 0.971 |
| Mean | 0.933 | 0.934 | 0.933 | 0.933 | 0.993 | 0.974 | 0.934 | 0.935 | 0.934 | 0.934 | 0.993 | 0.972 |
| **Class** | **Roberta Base** | | | | | | **Robert Base + CNN** | | | | | |
| Joy (0) | 0.928 | 0.93 | 0.927 | 0.928 | 0.993 | 0.972 | 0.927 | 0.931 | 0.922 | 0.926 | 0.993 | 0.971 |
| Fear (1) | 0.93 | 0.932 | 0.928 | 0.93 | 0.992 | 0.97 | 0.929 | 0.927 | 0.93 | 0.928 | 0.991 | 0.969 |
| Anger (2) | 0.931 | 0.929 | 0.93 | 0.929 | 0.993 | 0.969 | 0.93 | 0.93 | 0.928 | 0.929 | 0.992 | 0.971 |
| Sadness (3) | 0.927 | 0.928 | 0.932 | 0.928 | 0.992 | 0.971 | 0.928 | 0.928 | 0.927 | 0.928 | 0.992 | 0.97 |
| Disgust (4) | 0.929 | 0.933 | 0.929 | 0.931 | 0.993 | 0.973 | 0.926 | 0.928 | 0.926 | 0.927 | 0.992 | 0.97 |
| Shame (5) | 0.932 | 0.93 | 0.933 | 0.931 | 0.993 | 0.974 | 0.928 | 0.93 | 0.927 | 0.929 | 0.993 | 0.972 |
| Guilt (6) | 0.929 | 0.932 | 0.929 | 0.93 | 0.993 | 0.971 | 0.931 | 0.93 | 0.928 | 0.929 | 0.991 | 0.969 |
| Mean | 0.929 | 0.931 | 0.929 | 0.929 | 0.992 | 0.971 | 0.928 | 0.929 | 0.928 | 0.928 | 0.992 | 0.97 |
| **Class** | **Electra Base** | | | | | | **Electra Base + CNN** | | | | | |
| Joy (0) | 0.932 | 0.935 | 0.931 | 0.933 | 0.993 | 0.978 | 0.928 | 0.931 | 0.927 | 0.929 | 0.991 | 0.969 |
| Fear (1) | 0.934 | 0.936 | 0.932 | 0.934 | 0.994 | 0.976 | 0.93 | 0.932 | 0.928 | 0.93 | 0.992 | 0.967 |
| Anger (2) | 0.935 | 0.933 | 0.935 | 0.933 | 0.993 | 0.977 | 0.932 | 0.929 | 0.931 | 0.929 | 0.992 | 0.968 |
| Sadness (3) | 0.931 | 0.932 | 0.936 | 0.932 | 0.992 | 0.978 | 0.927 | 0.928 | 0.931 | 0.928 | 0.991 | 0.97 |
| Disgust (4) | 0.934 | 0.937 | 0.933 | 0.935 | 0.994 | 0.979 | 0.929 | 0.933 | 0.929 | 0.931 | 0.992 | 0.969 |
| Shame (5) | 0.936 | 0.934 | 0.937 | 0.935 | 0.993 | 0.976 | 0.931 | 0.93 | 0.932 | 0.931 | 0.992 | 0.967 |
| Guilt (6) | 0.933 | 0.936 | 0.933 | 0.934 | 0.992 | 0.978 | 0.93 | 0.932 | 0.929 | 0.93 | 0.992 | 0.969 |
| Mean | 0.933 | 0.934 | 0.933 | 0.933 | 0.993 | 0.977 | 0.929 | 0.93 | 0.929 | 0.929 | 0.991 | 0.968 |
| **Class** | **T5 Base** | | | | | | **T5 Base + CNN** | | | | | |
| Joy (0) | 0.754 | 0.756 | 0.753 | 0.751 | 0.952 | 0.831 | 0.732 | 0.74 | 0.731 | 0.73 | 0.948 | 0.808 |
| Fear (1) | 0.756 | 0.755 | 0.757 | 0.754 | 0.951 | 0.832 | 0.735 | 0.738 | 0.733 | 0.732 | 0.945 | 0.807 |
| Anger (2) | 0.758 | 0.753 | 0.755 | 0.753 | 0.952 | 0.833 | 0.738 | 0.736 | 0.735 | 0.733 | 0.947 | 0.805 |
| Sadness (3) | 0.755 | 0.752 | 0.759 | 0.752 | 0.951 | 0.834 | 0.731 | 0.735 | 0.736 | 0.729 | 0.946 | 0.804 |
| Disgust (4) | 0.757 | 0.759 | 0.756 | 0.755 | 0.952 | 0.833 | 0.734 | 0.739 | 0.734 | 0.73 | 0.944 | 0.806 |
| Shame (5) | 0.759 | 0.754 | 0.76 | 0.756 | 0.952 | 0.832 | 0.736 | 0.734 | 0.737 | 0.733 | 0.949 | 0.809 |
| Guilt (6) | 0.756 | 0.758 | 0.754 | 0.753 | 0.951 | 0.832 | 0.733 | 0.741 | 0.733 | 0.734 | 0.944 | 0.805 |
| Mean | 0.69 | 0.689 | 0.69 | 0.687 | 0.939 | 0.809 | 0.731 | 0.732 | 0.731 | 0.729 | 0.95 | 0.811 |

Electra Base + CNN and RoBERTa Base + CNN, especially in precision and recall. T5 Base + CNN fails to reveal much improvement on top of baseline T5, with barely marginal changes across metrics.

**Table 3 Results of the emotion classification DeBERT V3 base back translation, DeBERTa Base + CNN back translation, DeBERTa Base, and DeBERTa Base + CNN.**

| Class | Accuracy | Precision | Recall | F1 score | ROC AUC | PR AUC | Accuracy | Precision | Recall | F1 score | ROC AUC | PR AUC |
|---|---|---|---|---|---|---|---|---|---|---|---|---|
| | DeBERT V3 base back translation | | | | | | DeBERTa Base + CNN back translation | | | | | |
| Joy (0) | 0.803 | 0.344 | 0.397 | 0.329 | 0.707 | 0.415 | 0.886 | 0.558 | 0.567 | 0.56 | 0.851 | 0.601 |
| Fear (1) | 0.812 | 0.356 | 0.45 | 0.367 | 0.734 | 0.47 | 0.908 | 0.648 | 0.588 | 0.613 | 0.872 | 0.678 |
| Anger (2) | 0.891 | 0.377 | 0.374 | 0.375 | 0.741 | 0.491 | 0.909 | 0.631 | 0.674 | 0.649 | 0.885 | 0.731 |
| Sadness (3) | 0.793 | 0.295 | 0.408 | 0.316 | 0.693 | 0.405 | 0.885 | 0.558 | 0.526 | 0.539 | 0.843 | 0.612 |
| Disgust (4) | 0.898 | 0.396 | 0.397 | 0.396 | 0.718 | 0.505 | 0.858 | 0.728 | 0.813 | 0.734 | 0.895 | 0.772 |
| Shame (5) | 0.77 | 0.478 | 0.659 | 0.505 | 0.756 | 0.568 | 0.956 | 0.812 | 0.794 | 0.802 | 0.942 | 0.858 |
| Guilt (6) | 0.885 | 0.351 | 0.311 | 0.327 | 0.7 | 0.421 | 0.9 | 0.588 | 0.606 | 0.596 | 0.865 | 0.642 |
| Mean | 0.427 | 0.371 | 0.427 | 0.405 | 0.721 | 0.476 | 0.653 | 0.707 | 0.653 | 0.642 | 0.879 | 0.678 |
| Class | DeBERTa Base | | | | | | DeBERTa Base + CNN | | | | | |
| Joy (0) | 0.98 | 0.931 | 0.935 | 0.932 | 0.994 | 0.976 | 0.977 | 0.929 | 0.918 | 0.922 | 0.988 | 0.966 |
| Fear (1) | 0.983 | 0.943 | 0.937 | 0.94 | 0.995 | 0.974 | 0.979 | 0.931 | 0.926 | 0.928 | 0.991 | 0.966 |
| Anger (2) | 0.983 | 0.942 | 0.945 | 0.943 | 0.994 | 0.98 | 0.981 | 0.936 | 0.937 | 0.936 | 0.991 | 0.972 |
| Sadness (3) | 0.976 | 0.921 | 0.917 | 0.919 | 0.989 | 0.965 | 0.974 | 0.898 | 0.924 | 0.911 | 0.987 | 0.954 |
| Disgust (4) | 0.988 | 0.959 | 0.96 | 0.959 | 0.995 | 0.987 | 0.987 | 0.951 | 0.961 | 0.956 | 0.994 | 0.983 |
| Shame (5) | 0.992 | 0.975 | 0.974 | 0.974 | 0.998 | 0.993 | 0.992 | 0.978 | 0.968 | 0.973 | 0.997 | 0.993 |
| Guilt (6) | 0.981 | 0.935 | 0.932 | 0.933 | 0.993 | 0.972 | 0.98 | 0.94 | 0.922 | 0.93 | 0.991 | 0.964 |
| Mean | 0.943 | 0.9446 | 0.943 | 0.943 | 0.994 | 0.973 | 0.937 | 0.938 | 0.937 | 0.937 | 0.991 | 0.966 |

Performance measures for individual emotion classes demonstrate stability throughout most classes using XLNet, RoBERTa, Electra, and Disgust and Shame exhibit slightly superior performance in recall and F1 score assessments. The categorization of emotions class performed by T5 demonstrates a noticeable weakness because the model faces significant challenges with feelings of sadness, shame, and guilt. The dataset likely presents significant challenges when differentiating certain emotional states, leading to performance issues for classifiers. Generally, the transformer-based models XLNet, RoBERTa, and Electra perform well, especially XLNet, which performs better than the others. T5, however, performs poorly in emotion classification, which implies that it might not be as appropriate for multi-class emotion detection tasks. Adding CNN layers improves the performance of all models except T5, meaning that T5 may not gain as much from convolutional improvements compared to the other transformer models.

Table 3 displays the results of the emotion classification of the DeBERTa V3 Base and DeBERTa Base + CNN models that applied back translation techniques on seven emotion classes (Joy, Fear, Anger, Sadness, Disgust, Shame, Guilt) through their assessment of accuracy, precision, recall and F1 score, ROC AUC and PR AUC. Combining DeBERTa V3 Base and DeBERTa Base + CNN with back translation improves emotion classification results. The baseline accuracy of DeBERTa V3 Base, at 0.427, is adequate, but the model shows low precision and recall, specifically in the Joy and Sadness categories. Every fifth prediction of emotion since Joy belongs to the class tag was actually correct, but only one in

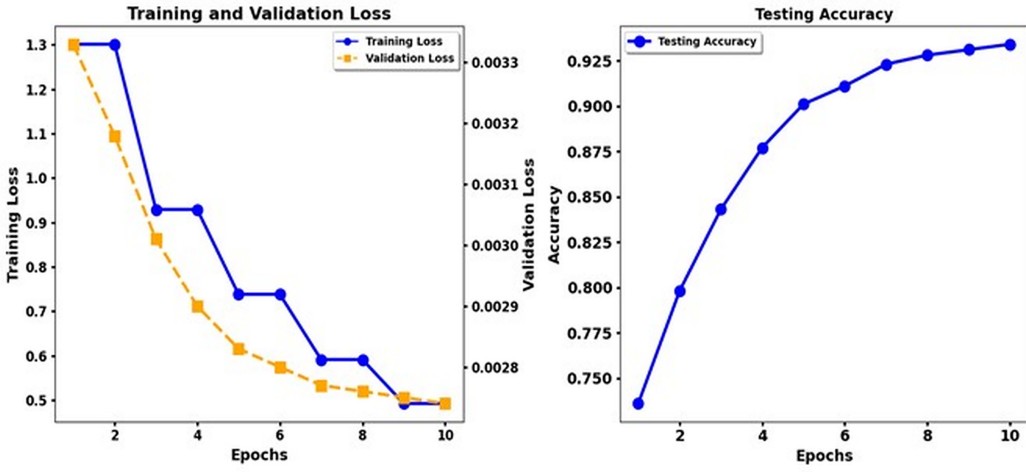

**Figure 1** Training loss and test accuracy: XLNET Base.

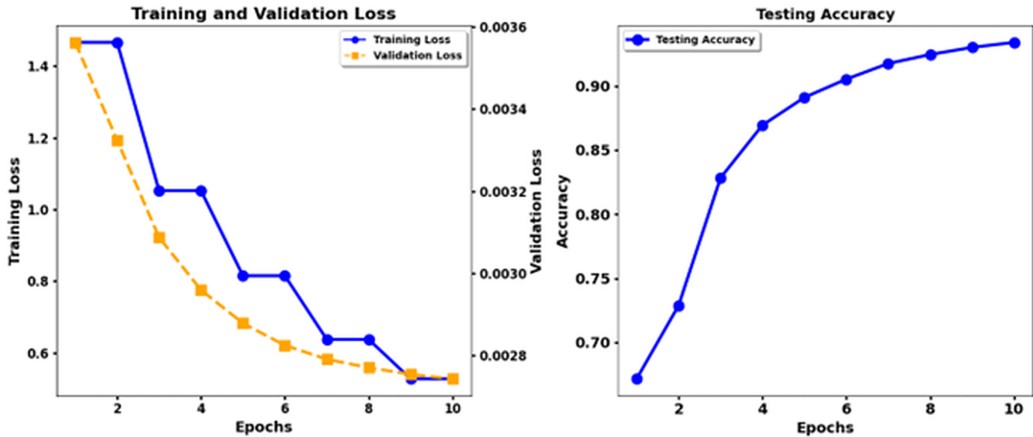

**Figure 2** Training loss and test accuracy: XLNET Base + CNN.

three instances was actually correct. Furthermore, Joy has an overall classification accuracy of 0.9875. Performance metrics, including the F1 score and the ROC AUC scores, show similar deficiencies, highlighting how challenging it is for the model to recognize emotions when dealing with class imbalance problems. A substantial improvement in performance emerges when CNN layers are connected to the DeBERTa Base models. The precision reached 0.653 after this enhancement, while Shame achieved a precision level of 0.956, and individual classes showed significant improvement. The models show enhanced precision and recall measurements for Fear and Shame while using CNN additions. These modifications better capture local dependencies, extracting more valuable features from text content. The F1 score increases for every class; therefore, Shame reaches an outstanding F1 score level of 0.802. The improvement in ROC AUC and PR AUC scores indicates that the model develops a higher ability to differentiate emotions. The study

shows that adding CNN layers improves the model's performance and accuracy in emotion detection tasks. Figure 1 shows the training loss and test accuracy of XLNET BASE, while Fig. 2 shows the training loss and test accuracy for the XLNET BASE + CNN model.

## DISCUSSIONS

The research examines the use of data augmentation and fine-tuning approaches to enhance emotion classification results through analysis of data from the ISEAR dataset. Different state-of-the-art models such as XLNet, RoBERTa, Electra, and T5 were evaluated using several metrics, including precision, recall, and the F1 score, in addition to the evaluation of the ROC AUC and PR AUC. Figure 3 shows the training loss and test accuracy of RoBerta-Base, while Fig. 4 shows the training loss and test accuracy for the Robert Base + CNN model.

Figure 5 shows the training loss and test accuracy of the Electra Base, while Fig. 6 shows the training loss and test accuracy for the Electra Base + CNN. Similarly, Fig. 7 shows the loss of training and the test precision of the T5 base, while Fig. 8 shows the loss of training and the test accuracy for the T5 base + CNN.

According to the results, the Transformer models XLNet and Electra demonstrate better performance than T5. The results show that XLNet Base delivers the highest performance, achieving 93. 3% precision, and Electra Base displays the matching results. T5 Base performs significantly behind other models in detecting emotional nuances because it achieves a low accuracy rate of 69%. The insertion of CNN enhancements into the XLNet Base + CNN, Electra Base + CNN, and Roberta Base + CNN systems achieves better precision and recall measurement performance because of their ability to extract local features. The combination of T5 Base + CNN does not deliver significant enhancements beyond T5 Base alone, making it appear that the T5 models lack the advantages of CNN layers. Recognition performance changes according to the different emotion categories. The model results indicated that disgust and shame performed best in most experimental scenarios, while the F1 score and the recall measures. The data set differentiates between feelings of Disgust and Shame more quickly, but the classifiers, particularly T5, struggled to distinguish between Sadness and Guilt. Performance analysis emphasizes why researchers need a detailed understanding of the different characteristics within emotional categories and how algorithms process these categories.

Figure 9 shows the training loss and test accuracy of the Debert V3 Base Back Translation, while Fig. 10 shows the training loss and test accuracy for the Deberta Base + CNN Back Translation. Similarly, Fig. 11 shows the loss of training and the test precision of the Deberta Base, while Fig. 12 shows the loss of training and the test precision for the Deberta Base + CNN.

Synonym replacement within data augmentation techniques produced a substantial performance gain because it expanded the training set while fixing insufficient data quantities and unbalanced classes. Expansion of the ISEAR dataset through these methods produced more emotions for models to learn, ultimately improving their ability to generalize beyond seen data. The research study endorses the suitability of transformer

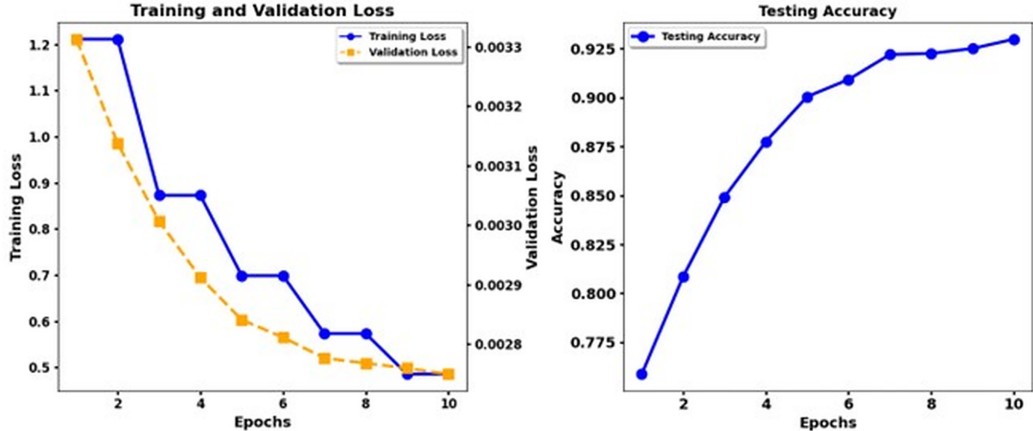

**Figure 3** Training loss and test accuracy: RoBerta-Base.

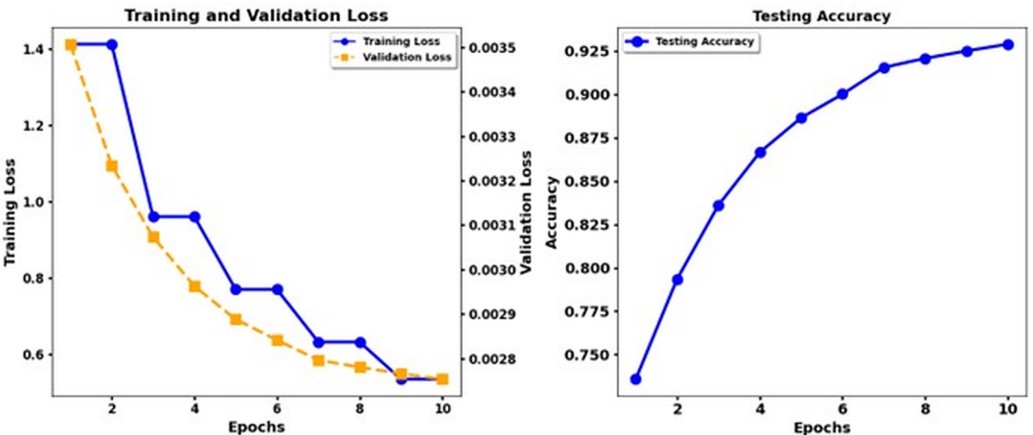

**Figure 4** Training loss and test accuracy: Robert Base + CNN.

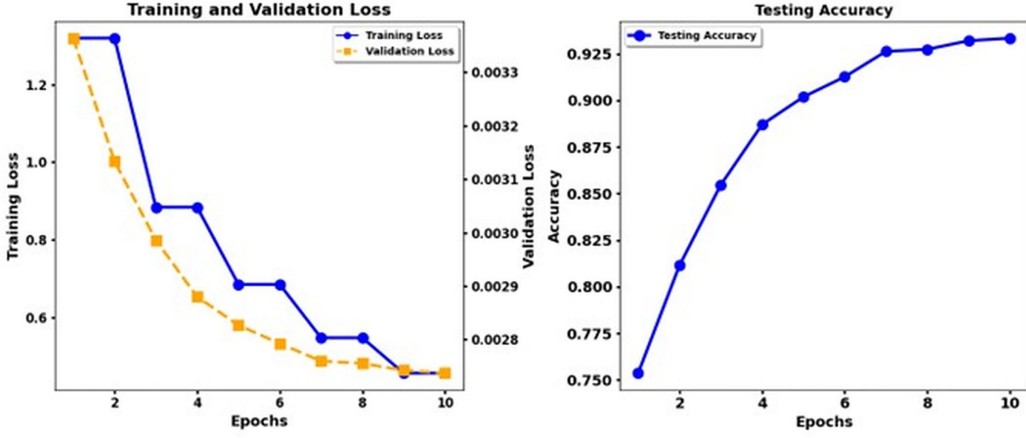

**Figure 5** Training loss and test accuracy: Electra Base.

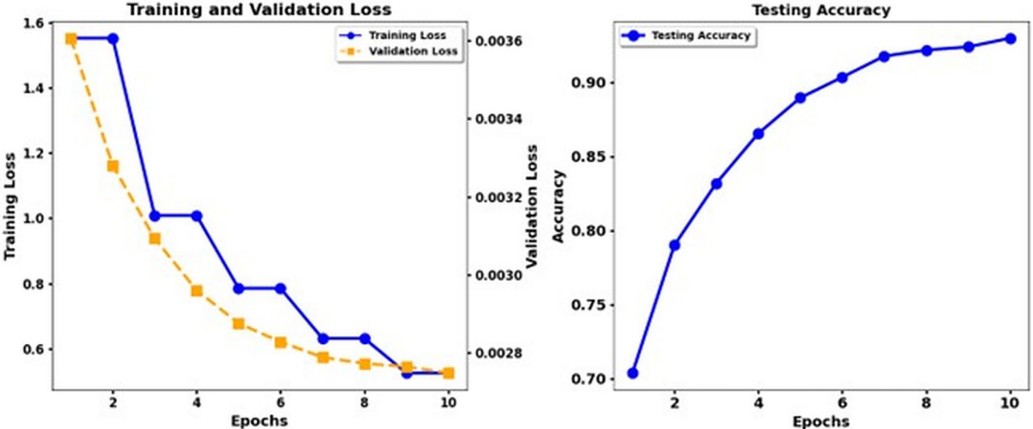

**Figure 6** Training loss and test accuracy: Electra Base + CNN.

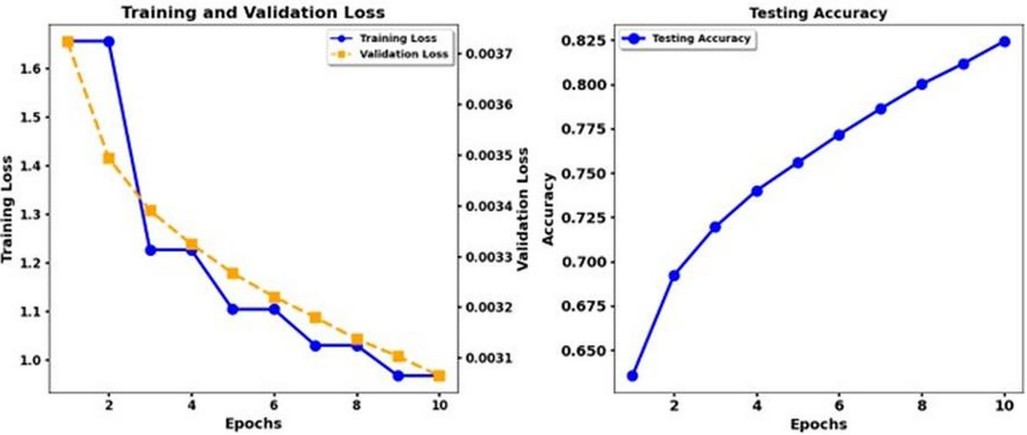

**Figure 7** Training loss and test accuracy: T5 Base.

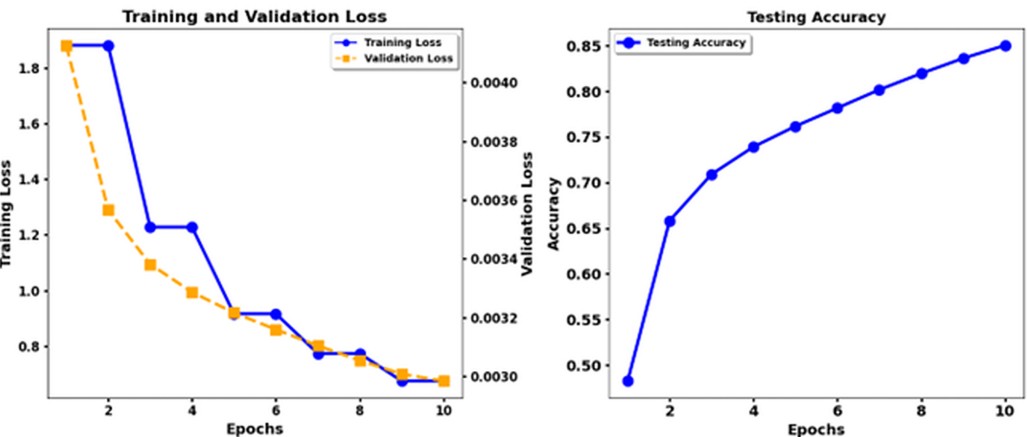

**Figure 8** Training loss and test accuracy: T5 Base + CNN.

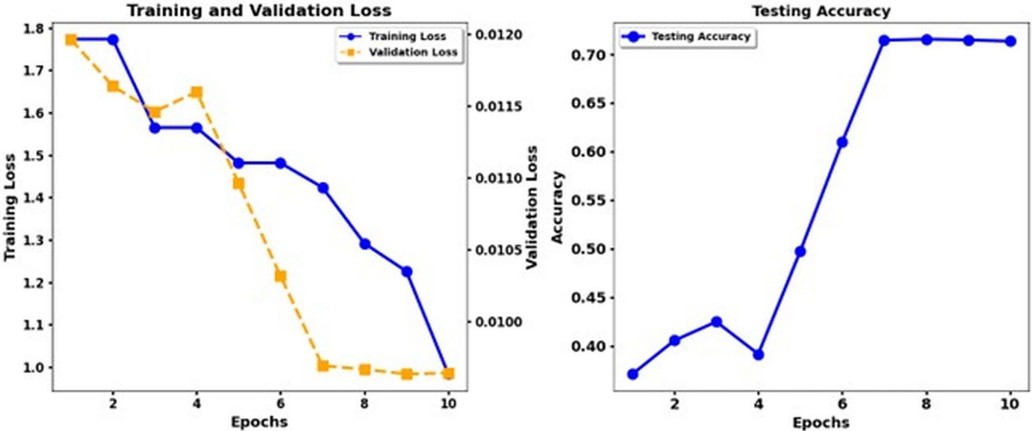

**Figure 9** Training loss and test accuracy: Debert V3 Base Back translation.

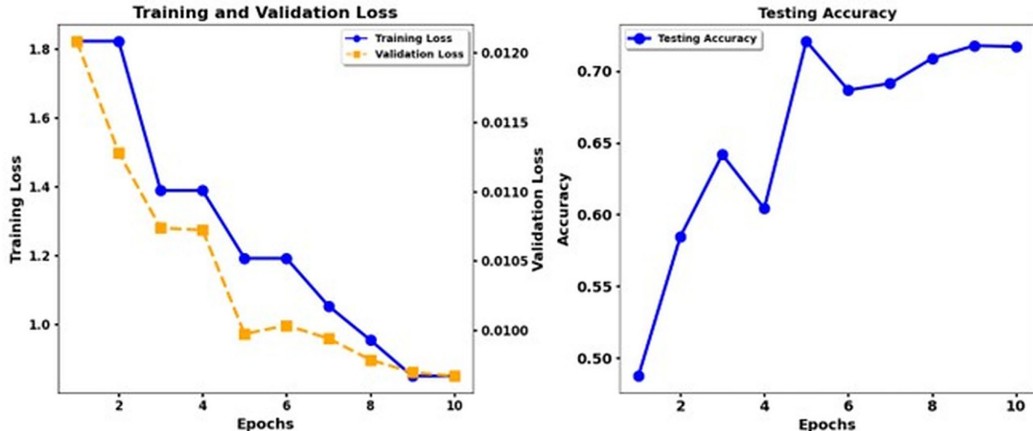

**Figure 10** Training loss and test accuracy: Deberta Base + CNN Back translation.

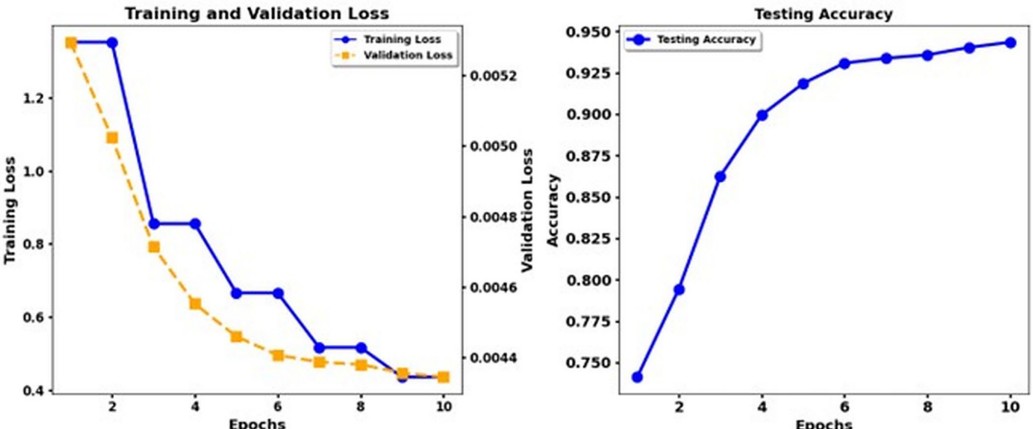

**Figure 11** Training loss and test accuracy: Deberta Base.

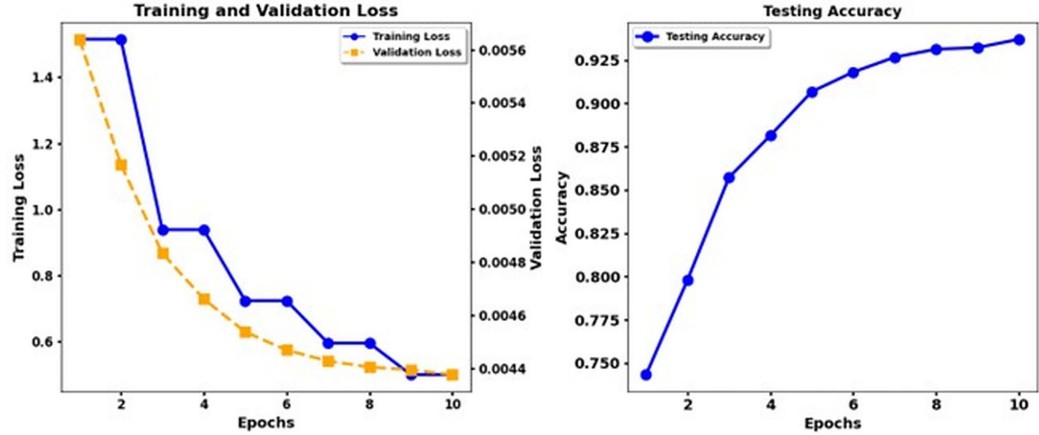

**Figure 12** **Training loss and test accuracy: Deberta Base + CNN.**

models, particularly XLNet when applied to emotion classification functions while showing positive effects from augmenting CNN pipelining methods. The findings indicate that T5 shows unsatisfactory results in detecting multiple emotions, which implies that this model should not be used for such tasks. Future research should develop more advanced augmentation approaches that include back-translation together with state-of-the-art technological methods like generative adversarial networks (GANs) to maximize model performance for complex emotional datasets.

The performance of the DeBERTa Base model continuously improves during ten epochs of training. Training loss exhibits a steady reduction throughout each model, and validation loss drops more slowly. The accuracy of the test develops gradually, but the DeBERTa Base model exhibits faster and smoother enhancement of the loss and accuracy functions until the accuracy increases above 0.9 in the 10th epoch. The DeBERTa Base model reaches high-performance standards similar to those of the third and fourth models while demonstrating consistent accuracy patterns and practical learning skills that demonstrate its strong generalizability.

## IMPACT OF CNN INTEGRATION

Multiple transformer-based models, including XLNet, Roberta, Electra, and T5, achieve better performance evaluation when CNN is integrated into these models. The addition of CNN to XLNet Base produces stable enhancements in performance metrics measured through accuracy, precision, recall, and F1 score values, which led to a mean accuracy growth from 0.933 to 0.934 and an increase in PR AUC values from 0.974 to 0.972. Roberta Base + CNN minimally enhances the model performance metrics, increasing the accuracy to 0.928 and advancing the PR AUC score to 0.970. Including CNN with Electra Base produces better precision and recall metrics, particularly for Fear and Guilt, surpassing default Electra Base performance. Significant enhancements occur through the T5 Base + CNN model configuration. T5 Base presents moderately weak performance at 0.690 accuracy level and 0.809 PR AUC, but the CNN integration leads this model to achieve

0.731 accuracy and 0.811 PR AUC, demonstrating a significant enhancement of emotional classification capabilities. Combining CNN with transformer-based models improves their emotion classification strength for difficult classes, thus yielding better accuracy metrics and precision-recall metrics for every tested configuration.

## CONCLUSION

This research examines how data augmentation methods affect the identification of emotions through analysis of the ISEAR dataset. The combination of data synonym replacement methods with CNN models connected to transformer-based systems, including XLNet, Roberta, and Electra, resulted in notable improvements to accuracy levels, precision and recall measurements, and F1 scores, particularly regarding the identification of disgust and shame emotions. XLNet showed the best performance in coping with complex emotional patterns among all models tested. Current challenges persist because the ISEAR dataset remains relatively tiny and unbalanced, affecting the performance outcomes of created models. The poor results of the evaluations of the T5 model reinforced the essential nature of selecting appropriate models for the emotion classification tasks. Developing advanced augmentation methods incorporating back-translation and Generative Adversarial Networks (GANs) in combination with multimodal input will make emotion detection systems more resilient while achieving higher accuracy.

The research shows variability in model performance across emotion classes, limiting its generalizability beyond a narrow range of emotions. This study has practical limits, although researchers have encountered promising outcomes. The performance of the classification models deteriorated significantly when processing the small and unbalanced ISEAR data set, particularly to identify sadness and guilt. The augmentation techniques used in this research only involve synonym replacement techniques, yet they omit more complex methods, including back-translation and GANs. The T5 model exhibited inferior performance compared to transformer models, which indicated that different architectural structures may not work equally well for emotion classification. The future research agenda includes developing improved data augmentation mechanisms. It requires the development of mixed models that merge linguistic and contextual features and expand the ISEAR dataset with data from multiple modalities to support more sophisticated emotion recognition.

### Funding

This work was funded by the King Salman center For Disability Research through Research Grant No. KSRG-2024-375. The funders had no role in study design, data collection and analysis, decision to publish, or preparation of the manuscript.

## Grant Disclosures

The following grant information was disclosed by the authors:
King Salman center For Disability Research through Research: KSRG-2024-375.

## Competing Interests

The authors declare that they have no competing interests.

## Author Contributions

- Uzair Muhammad conceived and designed the experiments, performed the experiments, analyzed the data, performed the computation work, prepared figures and/or tables, authored or reviewed drafts of the article, and approved the final draft.
- Khalil Ullah conceived and designed the experiments, performed the experiments, analyzed the data, performed the computation work, prepared figures and/or tables, authored or reviewed drafts of the article, and approved the final draft.
- Ibrar Hussain conceived and designed the experiments, performed the experiments, analyzed the data, performed the computation work, prepared figures and/or tables, authored or reviewed drafts of the article, and approved the final draft.
- Sulaiman Almutairi conceived and designed the experiments, performed the experiments, analyzed the data, performed the computation work, prepared figures and/or tables, authored or reviewed drafts of the article, and approved the final draft.
- Ikram Syed conceived and designed the experiments, performed the experiments, analyzed the data, performed the computation work, prepared figures and/or tables, authored or reviewed drafts of the article, and approved the final draft.
- Mohammed Abohashrh conceived and designed the experiments, performed the experiments, analyzed the data, performed the computation work, prepared figures and/or tables, authored or reviewed drafts of the article, and approved the final draft.

## Data Availability

The ISEAR Dataset is available at Kaggle and Hugging Face:
- https://www.kaggle.com/datasets/faisalsanto007/isear-dataset.
- https://huggingface.co/datasets/gsri-18/ISEAR-dataset-complete.

The ISEAR code is available at GitHub and Zenodo:
- https://github.com/ezrauzair/ISEAR-Paper-Code.
- Uzair Muhammad. (2025). ezrauzair/ISEAR-Paper-Code: ISEAR CODE (v1.0.0).
Zenodo. https://doi.org/10.5281/zenodo.15259205.

## Supplemental Information

Supplemental information for this article can be found online at http://dx.doi.org/10.7717/peerj-cs.2984#supplemental-information.

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
