# Peer review of "Enhancing emotion classification on the ISEAR dataset using fine-tuning and data augmentation with hybrid transformer models"

_PeerJ Computer Science, doi:10.7717/peerj-cs.2984_

## Round 0.1 · original submission · Major Revisions

· Academic Editor

Major Revisions

The paper presents promising work on transformer-based emotion classification. However, several critical issues must be addressed to improve its scientific rigour, clarity, and overall impact. Please, carefully consider all concerns raised by the reviewers and implement their feedback to improve both the content and presentation. Given these substantial revisions, we recommend a major revision.

Reviewer 1 ·

Basic reporting

Overall, this paper tackles an important topic—emotion classification via transformers and data augmentation—and its structure is sound. However, the language occasionally contains grammatical errors and awkward sentence constructions that disrupt readability (e.g., “high accurate classification” in the Abstract). The manuscript would benefit greatly from a thorough proofreading or professional editing pass to ensure precision and clarity in its technical discussion.

Although the introduction provides a reasonable overview of NLP emotion classification challenges, it lacks a clear statement on why hybrid transformer models are the optimal solution. The literature review is comprehensive yet could use a more critical comparison of past methods and deeper references to psychological theories of emotion. Strengthening the rationale for model selection (e.g., why these transformers and not GPT or LSTMs) would significantly improve the paper’s coherence.

While it offers valuable insights, several shortcomings need attention:
1. Model Selection and CNN Integration
o The paper doesn’t fully explain why it chose certain transformers or why GPT-based models or ALBERT were excluded.
o CNN layers help DeBERTa, but not always for Electra or XLNet; the paper should discuss why CNN boosts performance in some cases and not others.
2. Data Augmentation Concerns
o Only synonym replacement is used, risking altered meanings and possible label noise.
o The manuscript doesn’t isolate how much of the performance gain comes from augmentation versus the choice of model. Exploring other techniques (e.g., back-translation, paraphrasing, or GANs) could broaden the findings.
3. Metrics and Statistical Validation
o While accuracy, precision, recall, and F1 are reported, the ISEAR dataset is imbalanced, so metrics like precision-recall AUC or ROC-AUC would be more informative.
o No significance testing (e.g., t-tests) is provided, making it unclear if small improvements are truly meaningful.
4. Dataset Limitations and Generalizability
o The paper focuses solely on ISEAR, a relatively small dataset. Expanding to larger emotion benchmarks (GoEmotions, MELD, etc.) would show whether these methods generalize.
o There’s insufficient discussion on potential overfitting—particularly if augmented data is also used for validation.
5. Presentation and Writing
o Some redundancy (dataset imbalance mentioned repeatedly) and grammar slip-ups (e.g., “high accurate classification”) detract from clarity.
o The conclusion restates results but lacks deeper takeaways or solid guidance on future directions. A more engaging ending would help.

Experimental design

The paper’s experimental framework—assessing multiple transformer models with CNN augmentation—is a solid approach for tackling emotion classification. That said, certain aspects of the design could be more transparent: the paper doesn’t thoroughly detail hyperparameter choices (e.g., learning rate, batch size), and it doesn’t isolate the specific impact of data augmentation. Adding a straightforward comparison of models with and without augmentation, plus conducting statistical tests, would help clarify whether observed improvements are truly significant or just chance.

Further, replicability could benefit from more detailed data preprocessing steps and justifications for chosen metrics (e.g., why not include ROC-AUC or MCC?). A deeper error analysis of class-wise performance would also strengthen the results. Enhancing transparency in these areas would greatly boost the study’s rigor and make the findings more convincing.

Validity of the findings

While this paper demonstrates that certain hybrid CNN-transformer setups can boost emotion classification accuracy—most notably with DeBERTa—several elements undermine the strength of its conclusions. It doesn’t clearly distinguish how much performance improvement is due to data augmentation versus the underlying model choice or hyperparameter tuning. Additionally, there’s little to no statistical testing to confirm that these gains are significant rather than a random artifact.

Moreover, the authors do not explicitly state the study’s originality in light of existing CNN-transformer research, nor do they explore whether these methods would scale to larger or more diverse datasets. The conclusion summarizes the results but could more directly acknowledge limitations, discuss real-world feasibility, and suggest future directions, such as evaluating bias in emotion classification or integrating more robust statistical validation.

Additional comments

Suggestions for Improvement
• Clarify why these models were selected and examine GPT-style or lighter models (e.g., DistilBERT).
• Compare with more augmentation techniques, and show an ablation study to assess the true impact of augmentation vs. model architecture.
• Use additional metrics suited for imbalanced datasets, and include significance tests to validate the reported gains.
• Demonstrate the approach on multiple emotion datasets and analyze where the models fail or confuse similar emotions.
• Streamline the paper’s flow, refine language, and end with a more substantive conclusion discussing next steps, ethical concerns, and real-world applications.

Overall, this work has promise: the efforts to handle data scarcity, test a variety of transformer models, and integrate CNN layers are worthwhile. Addressing the above issues will greatly improve its scientific rigor, clarity, and usefulness

Annotated reviews are not available for download in order to protect the identity of reviewers who chose to remain anonymous.

Reviewer 2 ·

Basic reporting

1. Clarity and use of language
The document is written in clear English, with unambiguous language.The sentence structure is adequate and facilitates understanding of the content in most sections.

a) I think that the entire section 6 lacks an adequate narrative and story, being too schematized.
b) Including a table of acronyms (acronym/extended name/description) could improve readability.

2. Introduction and context
The introduction provides an adequate context for the study, presenting the problem clearly and well-structured. The topic's relevance is justified, and the motivation for the study is highlighted.

Although relevant articles are included in the state of the art. I recommend considering the following articles from the year 2024:

a) Hashmi, E., Yayilgan, S.Y. & Shaikh, S. Augmenting sentiment prediction capabilities for code-mixed tweets with multilingual transformers. Soc. Netw. Anal. Min. 14, 86 (2024). https://doi.org/10.1007/s13278-024-01245-6

b) Chutia, T., Baruah, N. A review on emotion detection by using deep learning techniques. Artif Intell Rev 57, 203 (2024). https://doi.org/10.1007/s10462-024-10831-1

3. Structure and publication standards
The document follows PeerJ standards and the norms of the discipline. No significant deviations are observed that affect the clarity of the work. Some formats errors are described following:

a) Alignment problems observed on lines 244-245
b) The beginning of each section should contain a description of issues that it will include (section 6)
c) In Table 4, I recommend changing the abbreviations used for performance metrics and using the commonly used ones P:Precision; A:Acuracy; R:Recall;
d) Line 276: The URL to access the augmented dataset is not displayed correctly (I was able to access it through the supplementary information included in the PeeJ platform).

4. Definitions and formal rigor
a) The methodology needs to be improved, it lacks a clear sequence of activities to achieve the objective, it lacks narrative.

Experimental design

The article content aligns with the aims and scope of the PeerJ Jornal (AI applications) . However, some errors were found:

a) Table 3 has display problems, the Class cell should be combined.
b) Why is Table 3 in materials & methods section and not in Results section?
c) Including section 4.3 as a justification for the data augmentation process, that is, integrate it with section 4.2
d) The beginning of each section should contain a description of what it will include (section 6)
e) In Table 4 I recommend changing the diminutive used for performance metrics and using the commonly used ones P:Precision; A:Acuracy; R:Recall;
f) Line 256 to 273: Describings the characteristics and sizes of the datasets in tables (6.3 section) in the same way as in Table 2.

Validity of the findings

With the integration of CNN (transformer-based model + CNN) it is not possible to generalize improved emotion detection performance to all transformer-based models, because it was only tested for a subset, and only a subset of them showed significant improvements.

The experiments and evaluations are well performed. The Conclusion section describe limitations, and future directions. A larger set of transformer-based models should be included (example: BERT base, DistilBERT, ALBERT, mBERT, among others)

Additional comments

Methods and tools section is very summarized and schematized, there is no narrative. It needs to be improved.

---

## Round 0.2 · Minor Revisions

· Academic Editor

Minor Revisions

The authors have made the requested improvements to the manuscript in response to the reviewers' feedback. However, minor issues remain—see Reviewer 2’s comments—that should be addressed before the manuscript is suitable for publication. A minor revision is therefore recommended.

Reviewer 1 ·

Basic reporting

The authors have satisfactorily addressed all of my previous comments. I have no further remarks.

Experimental design

The authors have satisfactorily addressed all of my previous comments. I have no further remarks.

Validity of the findings

The authors have satisfactorily addressed all of my previous comments. I have no further remarks.

Additional comments

The authors have satisfactorily addressed all of my previous comments. I have no further remarks.

Reviewer 2 ·

Basic reporting

The authors incorporated the recommendations made in the first revision and although Hashmi, E., Yayilgan, S.Y. & Shaikh, S. were cited in the text, they were not included in Table I.
At this point in 2025, it should include more citations from 2024 and incorporate them into the text and Table I.
The narrative of the documebt has improved significantly.

Experimental design

The methods & materials are improved and explained with enough detail and resources (such as code, dataset, computational setup, and reproduction scripts) to allow replication.

The URL to code was fixed.

Validity of the findings

No-comments

Additional comments

No-comments

---

## Round 0.3 · accepted · Accept

· Academic Editor

Accept

The authors have addressed all the reviewer's comments. The manuscript is now suitable for publication.

The Section Editor suggested:

> The section titled '11 LIMITATIONS AND FUTURE WORK' can be integrated as the concluding paragraph within the Conclusion section

Reviewer 2 ·

Basic reporting

The authors incorporated the recommendations made in the first and second revisions. The narrative of the document has been improved significantly. The methods & materials are improved and explained with enough detail and resources (such as code, dataset, computational setup, and reproduction scripts) to allow replication. The document structure is according to PeerJ standards and rules.

Experimental design

No comments

Validity of the findings

No comments

Additional comments

No comments